# Influence of Physicochemical Characteristics of Neem Seeds (*Azadirachta indica* A. Juss) on Biodiesel Production

**DOI:** 10.3390/biom10040616

**Published:** 2020-04-17

**Authors:** Bakari Hamadou, Ruben Zieba Falama, Cedric Delattre, Guillaume Pierre, Pascal Dubessay, Philippe Michaud

**Affiliations:** 1Energy Research Laboratory, Renewable Energy Section (LRE/SENC), Institute for Geological and Mining Research (IRGM), Nlongkak Yaounde P.O. Box 4110, Cameroon; 2Department of the Renewable Energies, The National Advanced School of Engineering of Maroua, University of Maroua, Maroua P.O. Box 46, Cameroon; 3Department of Physics, Faculty of Sciences, University of Maroua, Maroua P.O. Box 814, Cameroon; 4CNRS, SIGMA Clermont, Institut Pascal, Université Clermont Auvergne, F-63000 Clermont-Ferrand, France

**Keywords:** biodiesel, neem seed, transesterification, sanitation, biofuel

## Abstract

The aim of this work is to study the influence of the physicochemical characteristics of neem seeds, according to their mass and oil content, on the production of biodiesel. After the physical characterization of the seeds and extraction of the oil (triglycerides), biodiesel was produced from crude neem seed oil by transesterification with ethanol in the presence of sodium hydroxide. This study shows that the physicochemical characteristics of these seeds vary according to the origin of the samples. The seeds from Zidim, with a mass average of 200 seeds evaluated at 141.36 g and an almond content of 40.70%, have better characteristics compared to those collected in the city of Maroua, with average values evaluated at 128.00 g and 36.05%, respectively. Almonds have an average lipid content of 53.98 and 56.75% for the Maroua and Zidim samples, respectively. This study also reveals that neem oil, by its physicochemical characteristics, has a satisfactory quality for a valorization in the production of biodiesel. However, its relatively high free fatty acid content is a major drawback, which leads to a low yield of biodiesel, evaluated on average at 89.02%, and requires a desacidification operation to improve this yield. The analysis of biodiesel indicates physicochemical characteristics close and comparable to those of petrodiesel, particularly in terms of calorific value, density, kinematic viscosity, acid value, evaluated at 41.00 MJ/kg, 0.803, 4.42 cSt, and 0.130 mg/g, respectively.

## 1. Introduction

Neem (*Azadirachta indica* A. Juss) is a tree of Indian origin that adapts well to poor soils and is supported in warm and dry climates [1]. It is present in all the dry and subtropical tropical areas of Oceania, Asia, Africa and America [2]. It was introduced in the northern part of Cameroon thanks to the reforestation strategy called “Operation Green Sahel” set up by the Cameroonian government in the 1970s to fight against desertification. In the town of Maroua, capital of the Far North region of the country, this species represents about 80% of the trees population. These plants generate huge quantities of biomass, mainly leaves and seeds, which are not recovered, but are often swept away as waste and dumped or simply burned, causing sanitation and pollution problems. Indeed neem starts fructification most commonly between the ages of 3–5 years and reaches the maximum production at 10 years that continues until 150–200 years [3]. With an annual production estimated at 30–50 kg for an adult tree, neem seeds contain almonds very rich in oil with contents up to 60% of its dry matter [1,3,4]. The fatty acid profile of the neem seed oil is characterized by a ratio of saturated/unsaturated fatty acids ranging from 0.53 to 0.54 [5]. Several studies characterizing the fatty acid profile of the neem seed oil report that fourteen fatty acids have been identified, four of which are in the majority. Oleic (predominant fatty acid) is a monounsaturated fatty acid representing 25 to 58% [5,6,7,8], palmitic and stearic which are saturated, and linoleic which is polyunsaturated. In addition, the global energy demand with a predominance of fossil fuel sources, the use of which leads not only to its depletion, but also, and above all to global warming, is ever-increasing and requires us to develop renewable and sustainable alternative energy sources. Therefore, an effective and viable way to promote this sanitation is the energy recovery of this derivative of neem as the recovery of neem seeds could be useful for the production of biodiesel by transesterification. Biodiesel, which is a “clean”, biodegradable, non-toxic and renewable biofuel, will not only reduce greenhouse gas emissions, but will also reduce the problem of energy security. To this end, it is therefore important to characterize this biomass in order to better control its energy potential and the impact of the variability of these characteristics according to the different geographical and climatic zones from which the seeds come. This work aims to study the variability of physicochemical characteristics of seeds according to their mass and oil content and to establish their influence on biodiesel production by transesterification.

## 2. Material and Methods

### 2.1. Collection and Pretreatment of Plant Material (Neem Seeds)

There is only one variety of neem (*Azadirachta indica* A. Juss) in Cameroon. The two lots of neem seeds samples used for this study come from two localities in the Far North region of Cameroon with different geographic and climatic characteristics. Samples of seeds were collected in three different districts of the city of Maroua (“Pitoaré”, “Djarengol” and “Domayo”) and samples from Zidim, a locality of the department of Mayo Tsanaga, located about 48 km from the city of Maroua. Three samples were collected at different locations in each of these two localities. These neem seeds were obtained during the abundant fruiting period in March 2018 and then cleaned, dried, and stored at room temperature, protected from light and moisture and then the studies were conducted in April 2018. Their moisture contents were 9.533 ± 0.089 and 9.470 ± 0.049% for the Maroua and Zidim samples, respectively. The trials were carried out on whole seeds, depulpated seeds, almonds, hulls and pulps. In order to perform a physicochemical characterization of whole seeds, the samples received a series of pretreatment operations. They were cleaned and then dewatered by dipping them in water. The depulped seeds were dried at room temperature and then peeled to separate the shell from the kernel as shown in Figure 1.

### 2.2. Physicochemical Characterization of Neem Seeds

The almond and shell contents were determined by weighing after manual dehulling, from three samples of 200 depulped seeds. The pulp and skin content were also determined by weighing three samples of 200 g of seeds, before and after the pulping operation. The dry matter was determined according to the method of AFNOR [9], the known mass samples were dried at 105 °C until a constant mass was obtained and the dry matter content, expressed as a percentage, was calculated after cooling in a desiccator, according to the following formula:(1)$$DM_{C}=\frac{M_{f}-T\;}{M_{i}-T}\times 100$$

In the above equation, *DM_C_* is the dry matter content (%), *M_f_* is the mass of the crucible and sample after drying (g), *M_i_* is the mass of the crucible and sample before drying (g), *T* is the empty crucible tare (g).

The moisture content of the samples (*M_C_*) was deducted from the value of the dry matter according to the following relation:(2)$$M_{C}=100-DM_{C}$$

The inorganic material was determined from the dry matter samples by electric oven incineration at 550 °C for four hours and its value was calculated by the following formula:(3)$$I_{M}=\frac{M_{f}-T\;}{M_{i}-T}\times 100$$
where *I_M_* is the rate of mineral matter compared to DM (%), *M_f_* is the mass of the crucible and sample after calcination (g), *M_i_* is the mass of the crucible and dry sample before calcination (g), *T* is the empty crucible tare (g).

The lipid content was determined by oil extraction of samples with the Soxhlet according to the method described by UICPA [10]. Hexane was used as the organic solvent and the duration of the extraction operation was 8 h. After this operation, the solvent was removed by evaporation and the oil was dried in an oven.

The lipid content (h) in percentage dry matter is expressed according to the formula:H = ((m_1_ − m_0_)/m_1_) * 100 * 100/*DM_C_*(4)
where m_1_ is the mass of the flask containing the fat after steaming, m_0_ is the mass of the empty balloon.

### 2.3. Neem Oil Extraction

The mechanical extraction method is carried out with hot water after grinding almonds to a fine grain size. The almonds were previously crushed using a CORONA Moulinex and the ground material obtained was sieved using a sieve of 2 mm mesh size. A few drops of hot water at 60 to 70 °C were added to the grind while stirring vigorously until a black color was obtained which was followed by a squeezing operation to separate the oil from the cake. The oil extraction rate was determined by the ratio of the amount of oil collected by the oil content of the almond milling sample introduced into the extraction unit.

### 2.4. Physicochemical Characterization of Extracted Neem Oil

The iodine value (I_I_) which indicates the degree of unsaturation of the fatty acids in the oil was determined by the Wijs reagent method described by UICPA [10].

The iodine number was calculated through the relation:I_I_ = 12.69 × T (V_O_ − V_1_)/m(5)

In Equation (5), V_O_ is the volume of 0.1 N thiosulfate solution used for the blank test, V_1_ is the volume of the thiosulfate solution used for the sample, T is the exact title of the thiosulfate solution, m is the mass in g of the test sample.

The saponification value (I**_S_**) is the number of milligrams of KOH (Potassium hydroxide) needed to saponify 1 g of fat [10]. It is determined using the colored indicator (phenolphthalein) according to the method described by [10]. The saponification value (I_S_) was calculated according to the formula:I_S_ = 56.1 × N (Vo − V1)/m(6)
where 56.1 is the molecular weight of KOH, N is the normality of the HCl solution, m is the mass in g of the test sample, Vo is the volume of HCl used for the blank test, V1 is the volume of HCl used for the test portion.

The acid value (I_A_) is the number of milligrams of potassium hydroxide required to neutralize the free fatty acids present in 1 g of material [10]. It was determined using the colored indicator (phenolphthalein) according to the method described by UICPA [10] as:I_A_ = 56.1 × N_KOH_ × (V0 − V1)/m(7)
where V0 is the volume of the standard KOH solution used for the blank test, V1 is the volume of the standard solution of KOH used for the sample, N_KOH_ is the normality of the KOH solution, m is the mass in grams of the test sample, 56.1 is the molar mass of KOH.

The peroxide value (I_P_) was determined by the colored indicator method using starch paste as a colored indicator [10].

The peroxide value expressed in milliequivalents of active oxygen per kg of fat is given by:I_P_ = 1000 × N (V0 − V1) / m(8)

In Equation (8), M is the mass in g of the test sample, Vo is the volume of the thiosulfate solution used for the blank test, V1 is the volume of the thiosulfate solution used for the sample, N is the exact normality of the sodium thiosulfate solution used.

The lower calorific value (PCI) was determined by calculation using the saponification value and iodine value according to the empirical relationship of Haidara [11] reported by Faye [2] as:PCI = 11,380 − I_I_ − 9.15 * I_S_(9)

The viscosity is evaluated at 40 °C using a type C Haake viscometer. The density of the oil samples was determined according to the AFNOR standard [9] at 25 °C by the weighing method and calculated according to the following formula:d = m/m_eau_(10)
where d is the oil density, m is the oil mass (g), and m_eau_ is the mass of the same volume of water (g). 

### 2.5. Transesterification

It is an ethanolysis reaction of esters, also called triglycerides, contained in neem oil in the presence of a basic catalyst, the NaOH in this case, at a moderate temperature, that leads to the formation of glycerol and ethyl monoesters. It is materialized according to the reaction illustrated in Figure 2.

The reaction conditions and parameters are optimally set based on literature results in order to investigate the potential of neem oil for biodiesel production. This synthesis was carried out using NaOH as a catalyst at 1% of the neem oil mass, an ethanol/oil molar ratio of 6:1. The temperature of the reaction was set at 70 °C, at atmospheric pressure and with mechanical stirring for four hours of the reaction time. The mass yield of the reaction was calculated according to Equation (11).
R = (m_b_/m_h_) × 100(11)

In the above equation, R is the reaction yield (%), m_b_ is the mass of biodiesel (g), m_h_ is the neem oil mass (g).

At the end of the reaction, the ethyl ester (biodiesel) is separated from the glycerol and the other by-products of the reaction, by static decantation using the separating funnels for 24 h as shown in Figure 3. The excess ethanol was distilled off from the biodiesel at a temperature of 90 °C. The biodiesel was then purified (residual glycerine, traces of catalyst, soap, etc.) by washing with hot water at 60 °C followed by static decantation for 24 h (Figure 3). The residual water from the washing operation was removed by drying the biodiesel by steaming at 140 °C for 20 min.

### 2.6. Physicochemical Characterization of Biodiesel

The characterization of the biodiesel produced consisted of determining its intrinsic properties as fuels and then comparing them to the physicochemical characteristics of neem crude oil as well as those of petrodiesel fixed by international standards. These physicochemical characteristics, in particular the density, the viscosity, the calorific value, and the acid value, have been determined according to the same method and under the same conditions as those of the physicochemical characterization of the crude neem oil.

The statistical analysis of all the data in this study were recorded in an Excel spreadsheet using Microsoft Office software version 2013 and the results are expressed as average ± standard deviation. The variation in the data collected and the statistical significance of the treatment effect were analyzed by analysis of variance. Statistical differences with a probability value less than 0.05 (*p* < 0.05) are considered significant. When the probability is greater than 0.05 (*p* > 0.05) the statistical differences are not significant.

## 3. Results and Discussions

### 3.1. Variability of Physicochemical Characteristics of Neem Seeds

Covered with a thin skin, the neem fruit consists of a pulp containing seeds with one to three almonds depending on the seed size considered. The physicochemical characteristics of the studied seed samples presented in Table 1 vary according to their origin.

An analysis of the results indicate that samples of neem seeds from the Zidim locality have better physical properties compared to those collected in Maroua town. With a mass average of 200 seeds evaluated at 141.36 g and an almond content of 40.70%, these characteristics are significantly higher compared to the samples collected at Maroua, evaluated at 128.00 g and 34.05% for the mass average of 200 seeds and the almond content, respectively. These seeds have almond contents slightly lower than that of the Senegalese neem seeds studied by Faye in 2010, which ranged from 51.97% to 52.32% depending on the origin of the samples [2]. This variability of characteristics could be explained not only by the variation in climatic conditions including temperature and rainfall [12], but mainly by the diversity of the type of soil and the genotype of the trees. In addition, the seeds studied have a pulp and skin content slightly higher than that of the kernel with rates estimated on average at 48.61% and 44.46% for the samples of neem seeds from Maroua and Zidim, respectively.

Table 2 presents the centesimal distribution of the main constituents of the neem seeds studied. It can be seen from Table 2 that the kernel concentrates most of the lipid reserves of the seed and its lipid content varies according to the seed’s origin. These rates are estimated on average at 53.97% and 56.74% for samples from Maroua and Zidim, respectively. This means that the sample from Zidim is 2.77% more than the one from Maroua. Difference can be mainly explicated by harvest period and climatic, edaphic influence. These results are slightly higher than those obtained by Sagoua [13] and Nitièma [1] who evaluated lipid contents of almonds at 50% and 45.1%, respectively. However, this content is much closer to that of Senegalese neem seed kernels, which is about 55.95 to 58.38% depending on the origin of the samples [2]. A similar study in India also indicates that neem seeds have an oil content of between 40% and 60% [4]. In addition, the kernel also has a slightly higher dry matter content compared to that of the hull and that of pulp and hides. However, the essential mineral matter of the seeds (4.4%) is evenly distributed between the hull (2.0%) and the almond (2.0%), which confirms the results of Faye [2].

### 3.2. Physicochemical Characteristics of Neem Oil

With a strong odor and a very bitter taste, the physicochemical characteristics of the neem oil vary slightly with the provenance of the samples. The main physicochemical properties of the samples studied compared to other oils are summarized in Table 3.

Table 3 shows that the physicochemical properties of the studied neem oil samples vary very little according to the provenance of the samples. The acid value, one of the intrinsic characteristics of the oil, which strongly influences the transesterification yield, is evaluated on average at 8.97 and 9.16 mg/g. These values are relatively high compared to those of other oils such as jatropha and sunflower, whose average range from 1.8 to 2.5 mg/g [16] and from 1.56 to 1.88 mg/g [17], respectively. Moreover, the acid number values of the samples studied are in contradiction with those of Banik et al. [16] who also conducted a similar study on neem seed oil and reported an acid value of 28.6 mg/g. The density, evaluated on average at 0.833 and 0.850 for the Maroua and Zidim samples, respectively, were in line with the results reported by Banik et al. [14] in 2018, who also conducted a similar study on neem seed oil. The density results were lower than those of the neem oil obtained by some other authors which varied between 0.912 and 0.965 [3,15]. However, the results are closer to that of Senegalese neem seeds studied by Faye [2] with an average value of 0.864. The kinematic viscosity of our oil samples, evaluated on average at 26.34 and 26.67 cSt for Maroua and Zidim, respectively, confirms the results of Sekhar et al. [14], who obtained values between 20.5 and 48.5. Moreover, the calorific values of the samples are close to those of the neem oils found in the literature varying between 32 and 40 MJ/kg [17]. The saponification values are comparable to those of the jatropha oil, evaluated between 196 and 208 mg/g [14], and the iodine values are close to that of the olive oil evaluated between 75 and 94 mg I_2_/100 g [15]. These values also confirm the results of Sagoua [13], which indicate that neem oil iodine and saponification values are evaluated on average in the range of 65 to 80 mg I/100 g and in the range 175 to 205 mg/g, respectively.

### 3.3. Transesterification

The basic transesterification reaction of neem oil resulted in an average yield of 89.08% and 88.97% for the Maroua and Zidim samples, respectively. These results confirm those of Sekhar [15] who also performed the transesterification of neem oil under similar conditions and obtained a yield of 88%. This transesterification yield of neem oil is relatively low compared to that obtained under these operating conditions by other oils, in particular sunflower and karanja oil, evaluated at 97% [18] and 98% [19], respectively. This relatively low yield is mainly explained by the high free fatty acid content of neem oil (acid value evaluated in the range of 8.97 to 9.16 mg/g) which causes a parasitic reaction of saponification and, indirectly, the transesterification reaction yield decreases. This situation can be solved by a pretreatment of desacidification of neem oil before transesterification, in order to eliminate free fatty acids and to avoid the formation of soap [20].

### 3.4. Physicochemical Characteristics of Biodiesel (Neem Oil Ethyl Ester)

The main physicochemical characteristics of biodiesel including density, viscosity, lower calorific value, acid value were determined. The values obtained were compared with those from other similar studies which used standard biodiesel and petrodiesel.

Table 4 presents the physicochemical characteristics of the neem oil ethyl ester obtained by the transesterification reaction of the studied neem oil, in comparison with other biodiesel and petrodiesel.

The results presented in Table 4 show that seed sample provenance does not significantly influence the physicochemical characteristics of biodiesel. The lower calorific value (PCI) of the samples, evaluated on average at 40.997 and 41.008 MJ/kg, is similar to that of the neem oil obtained by Sekhar [15] and Banik et al. [17], which ranged from 39.6 to 40.2 MJ/kg. However, it is slightly higher than that of the neem oils studied by [21,22] as well as that of standard biodiesel [25] which were evaluated at 39.81 and 38. 586 MJ/kg, respectively. This PCI of neem oil biodiesel is similar to that of soybean oil and *Chlorella protothecoides* micoalgae oil, estimated at 40 MJ/kg [24] and 41 MJ/kg [25], respectively, but slightly higher than that of palm oil and sunflower oil, evaluated at 34 MJ/kg [26]. Moreover, the viscosity of the biodiesel obtained, evaluated on average at 4.36 and 4.49 cSt is comparable to that of the petrodiesel which varied from 3.1 to 4.7 cSt [17,23,27]. This result indicates that the transesterification reaction made it possible to reduce the neem oils viscosity to a factor of about 6.04 and 5.93 for Maroua and Zidim samples, respectively, compared with those of the respective biodiesels. The density of biodiesel samples, estimated at 0.802 and 0.805 is slightly lower than that of the neem biodiesel studied by Sekhar [15], which ranged from 0.820 to 0.940 but approximates that of petrodiesel which was about 0.83 [15,23,24]. The iodine values (49.280–49.491 mg I_2_/100 g) and the acid values (0.131–0.130 mg/g) of our samples are similar to those of the methyl ester of neem oil studied by Tanwar et al. [28], which were evaluated at 49.12 mg I_2_/100 g and 0.12 mg/g for the iodine value and the acid value, respectively.

## 4. Conclusions

This study aimed at contributing to the research on the energy recovery of neem seeds, particularly in regards to using neem seed oil as a means of sanitation and prevention of greenhouse gas emissions. It appears, from this experimental study, that the physicochemical characteristics of seeds vary according to their origin. The kernel content of the seeds is evaluated on average at 34.05 and 40.70% for samples from Maroua and Zidim, respectively. The lipid content of almonds is evaluated on average at 53.98 and 56.75% depending on the origin of the samples. With acid values averaging 8.97 and 9.16 mg/g for the Maroua and Zidim samples, respectively, neem oil has a relatively high free fatty acid content, which influences negatively on the yield of the transesterification reaction, evaluated at 88.97 and 89.08%. A deacidification operation is therefore necessary in order to improve the efficiency of the transesterification reaction of neem oil and to consider energy recovery of this biomass on an industrial scale. This study has also shown that ethyl ester, the biodiesel obtained following the transesterification of neem oil, has physicochemical characteristics very similar and comparable to that of petrodiesel, particularly in terms of density, viscosity and lower calorific value. 

## Figures and Tables

**Figure 1 biomolecules-10-00616-f001:**
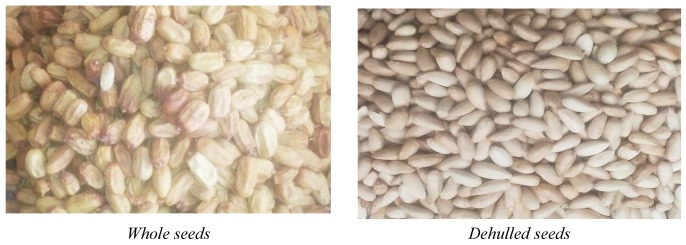
Images of the seed pretreatment operation.

**Figure 2 biomolecules-10-00616-f002:**
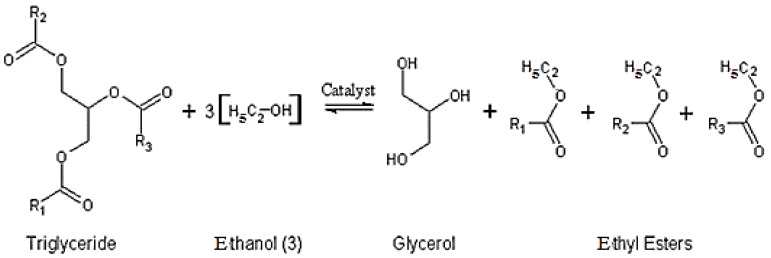
Transesterification reaction principle [12].

**Figure 3 biomolecules-10-00616-f003:**
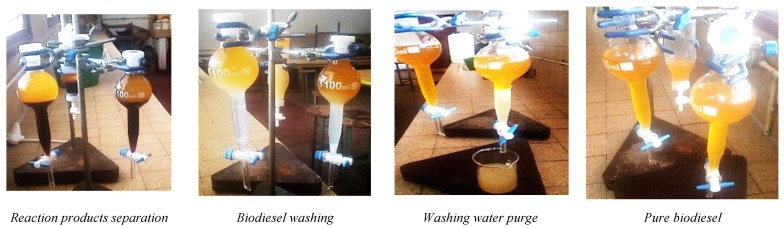
Illustration of biodiesel separation and purification operations.

**Table 1 biomolecules-10-00616-t001:** Physicochemical characteristics of neem seeds studied compared to those of Senegalese neem seeds.

Characteristics of Seeds	Neem Seeds from This Study		Neem Seeds [2]
	Maroua	Zidim	
Moisture content (%)	9.533 ± 0.089 ^a^	9.470 ± 0.049 ^a^	9.04–9.10
Average mass for 200 seeds (g)	128.000 ± 0.133 ^a^	141.367 ± 0.178 ^b^	237.20–268.50
Almond content (%)	34.050 ± 0.067 ^a^	40.700 ± 0.076 ^b^	51.97–52.32
Hull content (%)	17.317 ± 0.156 ^b^	14.817 ± 0.165 ^a^	47.68–48.03
Pulp and skin content (%)	48.617 ± 0.089 ^b^	44.467 ± 0.056 ^a^	-

Superscript letters indicate that statistical analysis doing by ANOVA, numbers with the same superscript letters on the same line indicate that these values are not significantly different at *p* < 5%.

**Table 2 biomolecules-10-00616-t002:** Percentage distribution of the main constituent elements of neem seeds.

Characteristics	Whole seed		Almond		Shell		Pulp and Skin	
	Maroua	Zidim	Maroua	Zidim	Maroua	Zidim	Maroua	Zidim
Dry Matter (%)	90.467 ± 0.089	90.530 ± 0.049	96.467 ± 0.177	96.333 ± 0.044	93.235 ± 0.326	93.333 ± 0.417	91.155 ± 0.031	91.217 ± 0.046
Mineral Matter (%)	4.422 ± 0.004	4.457 ± 0.049	2.073 ± 0.004	2.076 ± 0.002	2.050 ± 0.018	2.027 ± 0.022	0.609 ± 0.014	0.631 ± 0.011
Lipid (%)	31.926 ± 0.147	32.764 ± 0.367	53.978 ± 0.517	56.749 ± 0.357	-	-	-	-

**Table 3 biomolecules-10-00616-t003:** Physicochemical characteristics of the neem oils studied compared to those of other oils.

Physicochemical Characteristics of Oils	Neem Oil, this Study		Neem Oil [2]	Neem Oil [14]	Neem Oil [15]	Jatropha Oil [16]
	Maroua	Zidim				
Iodine value (mg I_2_/100 g)	74.448 ± 0.564	73.814 ± 0.366	75.93	65–80	-	89–95
Saponification value (mg/g)	200.090 ± 1.247	199.810 ± 1.584	199.17	175–20	-	196–208
Acide value (mg/g)	8.976 ± 0.610	9.163 ± 0.820	7.93	-	28.64	1.8–2.5
Peroxyde value (meq/Kg)	6.433 ± 0.151	6.900 ± 0.300	6.00	-	-	-
Lower Calorific value (MJ/Kg)	39.642	39.665	39.63	32–40	39.501	-
Density at 25 °C	0.833 ± 0.012	0.850 ± 0.014	0.86	0.912–0.965	0.93	0.895–0.902
Kinematic Viscosity at 40 °C (cSt)	26.34 ± 1.36	26.67 ± 1.57	-	20.5–48.5	40.7512	-

**Table 4 biomolecules-10-00616-t004:** Comparison of the biodiesel produced with other biodiesel, standard biodiesel and petrodiesel.

		Density at 25 °C	Acid Value (mg/g)	Kinematic Viscosity at 40 °C (cSt)	Iodine Value (mg I_2_/100 g)	Saponification Value (mg/g)	Lower Calorific Value (MJ/Kg)
Neem Biodiesel, this Study	Maroua	0.802 ± 0.023	0.131 ± 0.06	4.36 ± 0.51	49.280 ± 0.282	167.459 ± 0.561	40.997
	Zidim	0.805 ± 0.034	0.130 ± 0.03	4.49 ± 0.41	49.491 ± 0.635	167.365 ± 0.584	41.008
Biodiesel Standard (ASTM) [17]		0.88	0.80 max	1.9–6.0	-	-	38.586
Neem Biodiesel [21,22]		0.863	0.65	5.21	-	-	39.81
Neem Biodiesel [14]		0.875	0.8716	6.17	-	-	40.2
Neem Biodiesel [15]		0.820–0.940	-	3.2–10.7	-	-	39.6–40.2
Stone Fruit Oil Biodiesel [23]		0.855	0.25	4.26	104.7	-	39.64
Jatropha Biodiesel [22]		0.8795	0.4	4.8	104	-	39.23
Petro Diesel [15,23,24]		0.8272–0.83	0.05	3.1–4.7	38.3	-	42–45.30

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
