# Peer review of "Influence of Physicochemical Characteristics of Neem Seeds (Azadirachta indica A. Juss) on Biodiesel Production"

_biomolecules, 2020, doi:10.3390/biom10040616_

Round 1
Reviewer 1 Report
See attached file

Author Response
Dear reviewer,
Thank you for your comment.

Reviewer 2 Report
This manuscript presents the effect of physicochemical characteristics of Neem seeds (mainly mass and oil content) on the biodiesel production. This study is nicely organized and it will be interesting for readers of Biomolecules journal. The reviewer has the multiple queries listed below to improve the manuscript that should be replied in an exhaustive and sound basis before considering the article for publication.
1) The reviewer can see some errors in typos, grammatical errors, sentence structures, formatting errors, and font size in the whole manuscript that should be corrected.
2) Materials and methods parts need to be specific and provide more information that is detailed for someone who wants to follow this manuscript for a similar research. For example with the plant material, it should include year, month, water content and others.
3) Further research and comparison data are required in result section. The authors described several references in the manuscript; however, a high-level description with figures and table will improve the current study. The reviewer suggests that add a couple of Tables to compare the current work and previous studies with related references that will provide clear analysis.
4) The references in the work do not have recent research articles (1982 – 2012) that may not effectively present the key findings of the work. The reviewer suggests that add more recently published references and compare the data with the current work.
5) In result and conclusion sections, the authors explained the current work and tried to find a novelty of the current work. However, the reviewer thinks there is less to literature data regarding general and other physicochemical characteristics and other general/similar biomass materials for biodiesel production have been discussed. This type of information is of key importance, as it measure the relevance of this study.
Author Response
Point 1: The reviewer can see some errors in typos, grammatical errors, sentence structures, formatting errors, and font size in the whole manuscript that should be corrected.
Response 1: The whole manuscript has been completely re-read and corrections have been made.
Point 2: Materials and methods parts need to be specific and provide more information that is detailed for someone who wants to follow this manuscript for a similar research. For example with the plant material, it should include year, month, water content and others. 

Response 2: Details and additional information have been provided (line 61-75), marked in red.
Point 3: Further research and comparison data are required in result section. The authors described several references in the manuscript; however, a high-level description with figures and table will improve the current study. The reviewer suggests that add a couple of Tables to compare the current work and previous studies with related references that will provide clear analysis.
Response 3: Taking into account this suggestion, Table 1 (line 177-179); Table 3 (line 208-210) and Table 4 (line 247-248) were restructured, supplemented by data from similar studies while comparing the results of this study with those from the literature.
Point 4: The references in the work do not have recent research articles (1982 – 2012) that may not effectively present the key findings of the work. The reviewer suggests that add more recently published references and compare the data with the current work.
Response 4: This suggestion has been taken into consideration: the results of this study have been compared with those of much more recent studies conducted by some authors such as: Banik et al., in 2018 [16]; Anwar et al., in 2018 [26]; Anwar et al., in 2018 [27]; Hasni et al., in 2017 [22]…;
Point 5: In result and conclusion sections, the authors explained the current work and tried to find a novelty of the current work. However, the reviewer thinks there is less to literature data regarding general and other physicochemical characteristics and other general/similar biomass materials for biodiesel production have been discussed. This type of information is of key importance, as it measure the relevance of this study.
Response 5: Additional data were provided in both the Results section and the conclusion section (lines 215-220; 252-254; 258-263) to compare data from this study with those reported by authors in the literature.
Reviewer 3 Report
The manuscript ‘Influence of physicochemical characteristics of Neem seeds (Azadirachta indica A Juss) on biodiesel production’ should address the chemical characterization of neem seed oil from two region of Cameroon, and focus on their transesterification on biodiesel and their chemical characteristics. However, this paper doesn’t present any fresh information, overall the text is a mere listing of very common notions on biodiesel from seed oils. Detailed comments are addressed and major revision is advised.
1) I think sampling to be a fundamental part of an investigation process. The information here given is quite superficial. It is necessary to add a relevant detail. The studied organism(s) should be carefully identified and the voucher specimen number(s) should be reported. Reference specimens are essential data, not only for the identification of the plant material, but also because reference specimens represent the sample of the examined species which can be compared at any moment. Without these details the work loses all its scientific soundness. How many times and how many samples were collected for each species? How about the physiological conditions of plants (initial/advanced reproductive phase? Average weight of seeds?)? -The fruits were stored in fridge (what?) or immediately oven dried? These are important data in order to draw reliable conclusions on oil composition.
2) Materials & Methods
Line 62-123 : too many generalities mentioned in the text. Different formulas were already known.
3) Line 127-128 : Figure 2 illustrated a transesterification process with ethanol (EtOH) whereas in the abstract (Line 20), biodiesel was produced from crude neem seed oil by transesterification with methanol (MeOH). What about this ?
4) Line 130 : The litterature results were not mentioned… Provide the references.
5) Line 139 : The excess ethanol has been distilled off from the biodiesel at a temperature of 90 ° C. What about this temperature ? Is it not too high ?
6) Line 151-153 : Statistical analysis
What about the probability law - statistical test that was used? What is the confidence level ?
Results
7) Line 173-184: discussion about the differences in oil content is quite superficial. Oil content depends also on the physiological conditions of plants, on the process and solvent used… What about the extraction process on literature for comparison of the data ?
8) Authors should provide more detailed information on the oil fatty acids identification and composition (%) on neem seeds. I think FA composition to be a fundamental part of an investigation process for biodiesel production.
Author Response
Dear reviewer,
Thank you for your corrections and suggestions.
Please, fine response point by point
Point 1: I think sampling to be a fundamental part of an investigation process. The information here given is quite superficial. It is necessary to add a relevant detail. The studied organism(s) should be carefully identified and the voucher specimen number(s) should be reported. Reference specimens are essential data, not only for the identification of the plant material, but also because reference specimens represent the sample of the examined species which can be compared at any moment. Without these details the work loses all its scientific soundness. How many times and how many samples were collected for each species? How about the physiological conditions of plants (initial/advanced reproductive phase? Average weight of seeds?)? -The fruits were stored in fridge (what?) or immediately oven dried? These are important data in order to draw reliable conclusions on oil composition.
Response 1: This remark has been taken into account, and additional details and information have been provided. These descriptions were in lines 61-73.
Point 2: Materials & Methods
Line 62-123: too many generalities mentioned in the text. Different formulas were already known.
Response 2: Here no major modification has been made because this remark is contradictory to that of Reviewer 2, who suggests instead developing this part.
Point 3: Line 127-128: Figure 2 illustrated a transesterification process with ethanol (EtOH) whereas in the abstract (Line 20), biodiesel was produced from crude neem seed oil by transesterification with methanol (MeOH). What about this?
Response 3: This clarification has been made in the abstract section (line 20);
Point 4: Line 130: The litterature results were not mentioned… Provide the references.
Response 4: This change was made (line 144)
Point 5: Line 139: The excess ethanol has been distilled off from the biodiesel at a temperature of 90 °C. What about this temperature? Is it not too high?
Response 5: The objective is to eliminate the excess ethanol, but the evaporation temperature of ethanol being around 78.5 °C, which is lower than 90 °C. Heating at 90 °C simply allowed its rapid and complete evaporation without denaturing biodiesel.
Point 6: Line 151-153: Statistical analysis
What about the probability law - statistical test that was used? What is the confidence level?
Response 6: The answer to this concern was provided (lines 168-171).
Point 7: Line 173-184: discussion about the differences in oil content is quite superficial. Oil content depends also on the physiological conditions of plants, on the process and solvent used… What about the extraction process on literature for comparison of the data?
Response 7: The response to this concern was provided (line 193-200).
Point 8: Authors should provide more detailed information on the oil fatty acids identification and composition (%) on neem seeds. I think FA composition to be a fundamental part of an investigation process for biodiesel production.Response 8: The fatty acid composition of Neem seed oil was not determined in this study, however, the composition and fatty acid profile reported by several authors was mentioned in the introduction (line 45-49).
Round 2
Reviewer 1 Report
See report 2

Author Response
This study made it possible to carry out a comparative study of the physico-chemical characteristics of Cameroonian Neem seed oil to that of other regions of the world. It is clear from this study that: although there is a great similarity between some of their characteristics, there is a big difference in the acid number which is one of the most essential intrinsic characteristics which strongly influences the performance or the yield of the transesterification reaction. - the acid value of oils from Cameroonian Neem seeds is evaluated on average at 8.97 and 9.16 mg/g. «these acid number values of the samples studied are in contradiction with those of Banik et al. [16] who also conducted a similar study on Neem seed oils and reported an acid value of 28.64 mg/g.» (Line 213-219); - this study also made it possible to compare the characteristics of Neem seed oil with that of other oils, in particular jatropha oil, soybean oil, microalgae chlorella protothecoides oil, palm oil, sunflower oil ... (line 256-268; line 227-229) it was found that Neem oil has a relatively high acid number compared to other oils. And that it therefore requires a prior deacidification operation of this oil in order to improve the performance of the transesterification. (line 275-280).
Reviewer 2 Report
The revised version of manuscript is significantly improved; however, the current form is not acceptable for the publication. The reviewer has the multiple queries listed below to improve the manuscript that should be replied in an exhaustive and sound basis before considering the article for publication.
1. The reviewer still can see some minor errors in typos, formatting errors and different font size that should be corrected.
2. Introduction section is too broad to explain what are the main issues and problems and why this study with key findings is important to impact on the problems.
3. Figures 1-3 are not clear enough. Each figure has to have a high resolution and will be professional to be accepted to the journal.
4. Several references have added to the manuscript; however, the reviewer could not find the novelty of the current work compared to other published papers. There are some descriptions and comparisons with the references but the authors showed that the current work is similar to the previous studies but not presenting and addressing the key messages and novelty of the work.
Author Response
Point 1: The reviewer still can see some minor errors in typos, formatting errors and different
font size that should be corrected.
Response 1: The whole manuscript has been completely re-read and corrections have been
made.
Point 2: Introduction section is too broad to explain what are the main issues and problems
and why this study with key findings is important to impact on the problems. 

Response 2: Additional details have been provided for this purpose in the Introduction section
(line 56-58).
Point 3: Figures 1-3 are not clear enough. Each figure has to have a high resolution and will
be professional to be accepted to the journal.
Response 3: This suggestion has been taken into consideration: image quality improvement
treatments including resolution have been made in each of Figures 1 to 3.
Point 4: Several references have added to the manuscript; however, the reviewer could not
find the novelty of the current work compared to other published papers. There are some
descriptions and comparisons with the references but the authors showed that the current work
is similar to the previous studies but not presenting and addressing the key messages and
novelty of the work.
Response 4: This study made it possible to carry out a comparative study of the physico-chemical
characteristics of Cameroonian Neem seed oil to that of other regions of the world. It is clear
from this study that: although there is a great similarity between some of their characteristics,
there is a big difference in the acid number which is one of the most essential intrinsic
characteristics which strongly influences the performance or the yield of the transesterification
reaction.
- the acid value of oils from Cameroonian Neem seeds is evaluated on average at 8.97
and 9.16 mg/g. «these acid number values of the samples studied are in contradiction
with those of Banik et al. [16] who also conducted a similar study on Neem seed oils
and reported an acid value of 28.64 mg/g.» (Line 213-219);
- this study also made it possible to compare the characteristics of Neem seed oil with
that of other oils, in particular jatropha oil, soybean oil, microalgae chlorella
protothecoides oil, palm oil, sunflower oil ... (line 256-268; line 227-229) it was found
that Neem oil has a relatively high acid number compared to other oils. And that it
therefore requires a prior deacidification operation of this oil in order to improve the
performance of the transesterification. (line 275-280).

Reviewer 3 Report
The manuscript has been correctly revised. It could be accepted in present form.
Author Response

(The authors gave the same response as above.)
